

# The effects of mismatched train and test data cleaning pipelines on regression models: lessons for practice

James Nevin, Michael Lees and Paul Groth

Informatics Institute, University of Amsterdam, Amsterdam, Netherlands

## ABSTRACT

Data quality problems are present in all real-world, large-scale datasets. Each of these potential problems can be addressed in multiple ways through data cleaning. However, there is no single best data cleaning approach that always produces a perfect result, meaning that a choice needs to be made about which approach to use. At the same time, machine learning (ML) models are being trained and tested on these cleaned datasets, usually with one single data cleaning pipeline applied. In practice, however, data cleaning pipelines are updated regularly, often without retraining of production models. It is therefore common to apply different test (or production) data than the data on which the models were originally trained. The changes in these new test data and the data cleaning process applied can have potential ramifications for model performance. In this article, we show the impact that altering a data cleaning pipeline between the training and testing steps of an ML workflow can have. Through the fitting and evaluation of over 6,000 models, we find that mismatches between cleaning pipelines on training and test data can have a meaningful impact on regression model performance. Counter-intuitively, such mismatches can improve test set performance and potentially alter model selection choices.

## INTRODUCTION

In recent years, there has been an explosion in the use and study of machine learning (ML) models (*Aggarwal et al., 2022*). Such models have been applied to a number of different datasets to address a range of problems (*Libbrecht & Noble, 2015*; *Athey, 2018*; *Molina & Garip, 2019*). ML models are 'data-driven' because they are trained on specific data to complete a specific task, and their performance and characteristics are directly dependent on the data on which they are trained. However, the end-goal is always to use these models to perform some downstream task on different, unseen data. The implicit assumption is that the data on which these models are actually applied have the same distributions and characteristics as the data on which the models were trained.

*This assumption that the data at inference time matches the data used during training does not always hold in practice.* Frequently, data engineering and ML teams are distinct (*Dimensional Research, 2019*; *Friedel, 2023*), and perform their tasks semi-independently. Improvements and changes are made to data engineering pipelines that affect production

Corresponding author
James Nevin, j.g.nevin@uva.nl

data. ML models may not be fitted to this updated production data, as there are many costs associated with the retraining, retesting, and redeployment of the models to production (*Wiggers, 2019*; *Hecht, 2019*; *Paleyes, Urma & Lawrence, 2022*), with costs for ML projects performed by ITRex, for example, varying between \$26,000 and \$300,000 (*Shashkina, 2024*). As a result, the data on which these models are trained may not match the data on which they are applied in practice. The frequency of such mismatches is also exacerbated by the now common practice of reusing datasets (*Paullada et al., 2021*) and possible differing quality in training and test data (*Budach et al., 2022*). This is at odds with the ML research space, where the focus is on so-called 'benchmark datasets' that have perfectly matched training and test data. These benchmarks are limited in number and in many cases will not match the production data (*Suresh & Guttag, 2021*; *Groh, 2022*).

Given these potential mismatches between training and production data, it is important to account for where such differences can arise and what impact they can have on model performance. One important source of such discrepancies is through data cleaning pipelines. These pipelines aim to identify and correct data quality issues within datasets. Such corrections can have a potentially large impact on the data, and thus model behaviour. Historically, data quality and ML model fitting and selection have been done somewhat independently (*Li et al., 2021*). However, these processes can have significant interactions with each other (*Mazumder et al., 2022*; *Northcutt, Athalye & Mueller, 2021*; *Budach et al., 2022*).

In this article[1], we investigate the impact that mismatched data cleaning pipelines can have on ML models, specifically for regression tasks, for which the effects of data quality/data cleaning are generally less well-understood than for classification tasks (*Martín, Sáez & Corchado, 2021*). Using two datasets from *Li et al. (2021)*, we define a regression task on each. For each dataset, we create a collection of different data cleaning pipelines. Train and test sets are cleaned independently, leading to a number of cleaned versions of each dataset. For each cleaned version of a given train dataset, three different regression models are trained (Random Forest Regression (*Breiman, 2001*), Gradient Boosted regressor, and extreme gradient boosting (XGBoost) regressor (*Chen & Guestrin, 2016*)). These models are then tested on each of the cleaned test sets. These experiments lead us to identify a number of key findings:

1. The method used for cleaning the test data can be far more influential on the performance than the method for cleaning the training data.
2. The best performance is not necessarily achieved when the same cleaning process is applied in the test and training data.
3. Different cleaning pipelines for test data can change the best-performing model type for a given training cleaning pipeline.

The main contributions of this article are as follows:

- Identification of the need to test the effects of mismatches between training and test set cleaning pipelines. While the effects of changes in production data on ML model

[1] Portions of this text were previously published as part of a thesis (*Nevin, 2024*).

performance is of known importance, it is not often considered from the perspective of changes in cleaning pipelines.

- A number of experiments highlighting how such mismatches can affect the performance of regression models. Regression models are frequently under-emphasised in the literature, despite their prominence in real-world application.

In "Background" we introduce the relevant background work; "Experiments" describes the data, models, and cleaning setups used in the experiments; "Results" and "Discussion" present and discuss the results; finally, "Conclusions" offers general conclusions, limitations, and future work.

## BACKGROUND

The problem of data quality for ML models has long been studied (*Blake & Mangiameli, 2011*; *Gudivada, Apon & Ding, 2017*). *Li et al. (2021)* note that the issues of data quality and ML model performance are often addressed separately by the ML and databases communities. The authors collected a number of real-world datasets with different error types and rates and tested classical classification algorithms. They showed that there is a consistent impact from data cleaning with respect to model performance, and this can mean that data cleaning is generally more applicable than creating very robust models. *Mazumder et al. (2022)* identified limitations of current benchmarks in ML research and noted the importance of data quality in training ML models. *Northcutt, Athalye & Mueller (2021)* investigated how pervasive errors in labels can destabilise the ML benchmark. Many test data are mislabelled, and correcting these errors can change which models perform relatively better or worse. This highlights the need to improve the labelling of test data in order to better choose which models to use. *Budach et al. (2022)* showed how the performance of ML models can be affected by systematically introducing data quality errors into training data, highlighting the (lack of) robustness for some model-dataset combinations.

In addition to ML model performance, there is also an increasing emphasis on issues such as fairness and explainability with respect to data quality. *Zelaya (2019)* defines volatilities to measure the effects of preprocessing on different data points. They note that different data preprocessing can impact predictions of different data points in different ways, and this could be correlated to specific features. *Biswas & Rajan (2021)* note that all steps in the ML pipeline can cause unfairness, but the focus has thus far generally been on the models themselves. Many typical data preprocessing steps (missing value removal, data filtering, feature engineering, *etc.*) can introduce unfairness, and this effect can be positively or negatively related to model performance. *Guha et al. (2023)* shows how automatic data cleaning can hurt fairness, with issues such as missing values more common in disadvantaged groups.

The studies on data quality's effects on ML models have generally been limited to classification tasks (*Martín, Sáez & Corchado, 2021*), and there is a relatively high level of understanding of how classification models deal with noisy data. Algorithms such as XGBoost and Random Forests either have built-in methods for addressing noise or have

been shown empirically to deal with it well (*Gómez-Ríos, Luengo & Herrera, 2017*; *McDonald, Hand & Eckley, 2003*). Other tools like k-nearest neighbor (k-NN) (*Biau & Devroye, 2015*) and support vector machines (SVMs) (*Schölkopf, 2002*) have been shown to be sensitive to noise in labels (*Martín, Sáez & Corchado, 2021*). Most of the the literature on noise in regression tasks has been focused on SVMs (*Martín, Sáez & Corchado, 2021*; *Borah & Gupta, 2019*; *Cherkassky & Ma, 2004*; *Suykens et al., 2002*). Robustness for other algorithms such as Random Forests and gradient boosted models has been investigated to some extent (*Brence & Brown, 2006*; *Roy & Larocque, 2012*; *Einziger et al., 2019*), but these efforts are infrequent, and the emphasis is more on parameter tuning robustness rather than noise in the data. The performance of regression models is measured in a fundamentally different way than classification models: predictions are not only right or wrong, but instead have their accuracy expressed as a (potentially highly variable) numerical error. This difference could affect the sensitivities of performance metrics to data quality concerns, emphasising the need for dedicated study.

The aforementioned data quality concerns advocate for the necessity of regularly maintaining and updating data preprocessing pipelines as new information is received. However, these updates to preprocessing pipelines can cause shifts in the data, affecting the performance and applicability of an already-trained ML model. Retraining and redeploying an ML model has many costs associated with it, and these costs are rapidly growing for the most advanced artifical intelligence (AI) models (*Cottier et al., 2024*). Because of this, how best to address and manage these costs is an active area of research. *Mahadevan & Mathioudakis (2024)* introduce the concept of 'cost-aware' retraining for ML models, where a balance is struck between staleness costs (the costs of lower model performance on updated data) and retraining costs (money and time costs of retraining). This is also being considered in industry, where questions are regularly asked about how production data are changing (*Baier, Reimold & Kühl, 2020*; *Podkopaev & Ramdas, 2021*) and when retraining should be done (*EvidentlyAI, 2024*). Most of the thinking on this issue is on the effects of new data or changes in the distributions of the production data not as a result of preprocessing, but rather some other external factor.

All of the work described above is relatively recent, highlighting the increasing need to address these issues. We differentiate from this related work in a number of ways:

- We focus on the less well-understood regression models, rather than the more commonly studied classification models.
- We do not use synthetic errors, but rather real-world datasets.
- We compare mismatched training and test set cleaning pipelines, rather than simply cleaned *vs*. dirty training and test sets.

## EXPERIMENTS

We use two different datasets that have one or many data cleaning issues and were originally used for classification tasks, defining new regression tasks on them. These data (and the general approaches to cleaning) are from CleanML (*Li et al., 2021*) (https://github.

com/chu-data-lab/CleanML). The three types of errors addressed in these experiments are duplicates, missing values, and outliers. For each, different identification and correction strategies are defined and all possible combinations tested. This creates multiple cleaned training and test sets.

Three different types of regression model are trained: Random Forest regression (RFR), Gradient Boosted regressor (GBR), and XGBoost regressor (XGB). These three models were chosen because they strike a good balance in task performance, model complexity, and training times on the datasets, while not having been well-studied in the literature for data quality robustness. Models are fitted to the various cleaned training sets, and subsequently tested on different cleaned test sets.

In this article, we focus on scale in the number of data cleaning pipelines. As a result, we generally limit our analysis to simple models, data cleaning, and performance metrics. More sophisticated and nuanced approaches are possible and offer potential avenues for further research.

## Datasets

There is a general lack of real-world datasets with a number of non-synthetic, impactful errors (*Li et al., 2021*; *Neutatz et al., 2022*). Most potential datasets available with non-synthetic errors were defined for classification tasks, and do not have non-trivial regression tasks. As a result, we limited our testing to the two applicable datasets from *Li et al. (2021)* (https://doi.org/10.5281/zenodo.14945262), which are real-world data with non-synthetic data quality errors and meaningful regression tasks.

The first dataset tested is USCensus. This dataset includes a number of attributes on persons based on a census in the United States. Attributes include things such as income, education level, and more. The regression task is to predict the age of a given person. The data cleaning issue addressed in this dataset is missing values. The total dataset has 32,561 rows, of which 2,399 have missing values, *i.e.*, 7.4% of rows.

The second dataset tested is Airbnb. This dataset contains information on Airbnb rentals, such as the number of bedrooms/bathrooms, location, and review scores. The regression task is to predict the price of the Airbnb. The data cleaning issues considered for this dataset are missing values, outliers, and duplicates. There are a total of 33,145 rows in the full dataset. Of these, 6,857 (20.7%) have missing values. Duplicates are identified on the basis of the latitude and longitude of the Airbnb. There are a total of 3,429 (9.8%) duplicates in the data.

## Data cleaning

Data cleaning issues are identified and addressed using automatic, standard techniques. These techniques are relatively simple, but are often employed in practice (*Li et al., 2021*) and are fast to execute. The three issues are: *missing values*, *outliers*, and *duplicates*. Implementation of the techniques described below is done using a slightly altered version of the MVCleaner class from CleanML. This cleaning information is summarised in Table 1. Note that for the identification technique for missing values, 'NA' constitutes a placeholder.

**Table 1 Data cleaning issues: identification and correction strategies.** Missing value corrections are in the form 'Numerical Correction-Categorical Correction'.

| Data cleaning issue | Identification techniques | Correction techniques |
|---|---|---|
| Missing values (Numerical-Categorical) | NA | Delete, Mean-Mode, Median-Mode, Mode-Mode, Mean-Dummy, Median-Dummy, Mode-Dummy |
| Outliers | Ignore, SD, IQR | Mean, Median, Mode |
| Duplicates | Ignore, Key value | Delete |

Missing values are dealt with in two general ways: either any row with one or more missing values is deleted, or the missing values are filled using an 'average'. For numerical attributes, this average can be the mean, median, or mode of the non-missing values for that attribute. For categorical missing values, this average can be either the mode or a dummy ('MISSING') value. Missing values are always identified and corrected, since one of the algorithms tested does not support missing values.

Numerical outliers can be ignored or identified and corrected. If ignored, outliers are assumed to be undetected. Alternatively, outliers are identified in one of two ways: either through checking which values are more than three times the standard deviation (SD) from the mean, or 1.5 times the interquartile range above or below the upper and lower quartiles, respectively (IQR). Identified outliers are replaced with either the mean, median, or mode of the attribute.

Duplicate rows in the data can either be ignored/assumed undetected or identified based on some key value column(s). If duplicates are addressed and two or more rows have the same values for the key value column(s), all but one of these rows are deleted.

The same selected methods of identification and correction for all issues are applied to every attribute, *e.g.*, different attributes do not use different missing value correction strategies for a given cleaning pipeline.

## Regression models

In general, there is a far better understanding of how classification models are able to deal with low-quality data. While there is some active research for regression tasks, this is usually limited to certain regression models like SVMs and deep learning models (*Martín, Sáez & Corchado, 2021*; *Cui et al., 2023*). The regression models we test represent medium levels of complexity, perform well on the tasks, and are not well-understood in regard to low-quality data robustness.

The three model types tested are RFR, GBR, and XGB. The models are fitted using GridSearchCV from sklearn 1.3.2 (*Pedregosa et al., 2011*). Scores for fitting are calculated using the negative mean squared error; cross-validation splitting is done using the default 5-fold cross-validation. For RFR and GBR, the grid search is based on the minimum samples split and the number of estimators; for XGB, the learning rate, the number of estimators, and the maximum depth are used.

**Table 2 GridSearchCV parameters (USCensus).**

| Model | Minimum samples split | Number of estimators | Learning rate | Max depth |
|---|---|---|---|---|
| Random forest regression | [60, 70, 80] | [350, 400, 450] | NA | NA |
| Gradient boosted regressor | [8, 10, 12] | [650, 700, 750] | NA | NA |
| XGBoost regressor | NA | [700, 800, 900] | [0.04, 0.05, 0.06] | [3, 4, 5] |

**Table 3 GridSearchCV parameters (Airbnb).**

| Model | Minimum samples split | Number of estimators | Learning rate | Max depth |
|---|---|---|---|---|
| Random forest regression | [2, 3, 4] | [300, 325, 350] | NA | NA |
| Gradient boosted regressor | [3, 4, 5] | [100, 125, 150] | NA | NA |
| XGBoost regressor | NA | [700, 800, 900] | [0.07, 0.08, 0.09] | [5, 7, 9] |

For each dataset and model combination, the grid search parameters are chosen on the basis of the full dataset with missing values deleted and no other cleaning applied. This is to ensure a baseline consistency between the three models, as GBR does not support missing values. The best-performing grid on this full dataset is used for all subsequent model fittings for the given dataset and model pair.

The search parameters for the USCensus data are shown in Table 2, and for the Airbnb data in Table 3. For these experiments, the focus is on relative model performance based on data cleaning. As a result, we are less concerned with the exact model parameters used, how they were selected, *etc*.

## Experimental setup

For each dataset, an initial set of 20 different train-test splits (in a 70–30 split) from the full uncleaned data are created. Assuming $n$ different cleaning pipelines, the experiments proceed as follows for each of the train-test splits:

1. Both the training and test sets are cleaned independently using $n$ cleaning pipelines. This results in $n$ cleaned training sets and $n$ cleaned test sets
2. The three different models are each trained on all $n$ cleaned training sets, giving a total of $3n$ fitted models
3. The performance of all $3n$ models are tested on all cleaned test sets, for a total of $3n^2$ performance scores

For a given training cleaning pipeline and test cleaning pipeline, model performance is averaged over 20 different train-test splits. For these experiments, performance is measured using the $R^2$ score, which serves well as a general overview of model performance, highlights the differences that can be observed, and has already been used in testing model robustness (*Einziger et al., 2019*). Figure 1 shows the experimental procedure for a single train-test set and model type. The same train-test splits are used for all three model types (RFR, GBR, XGB) tested.

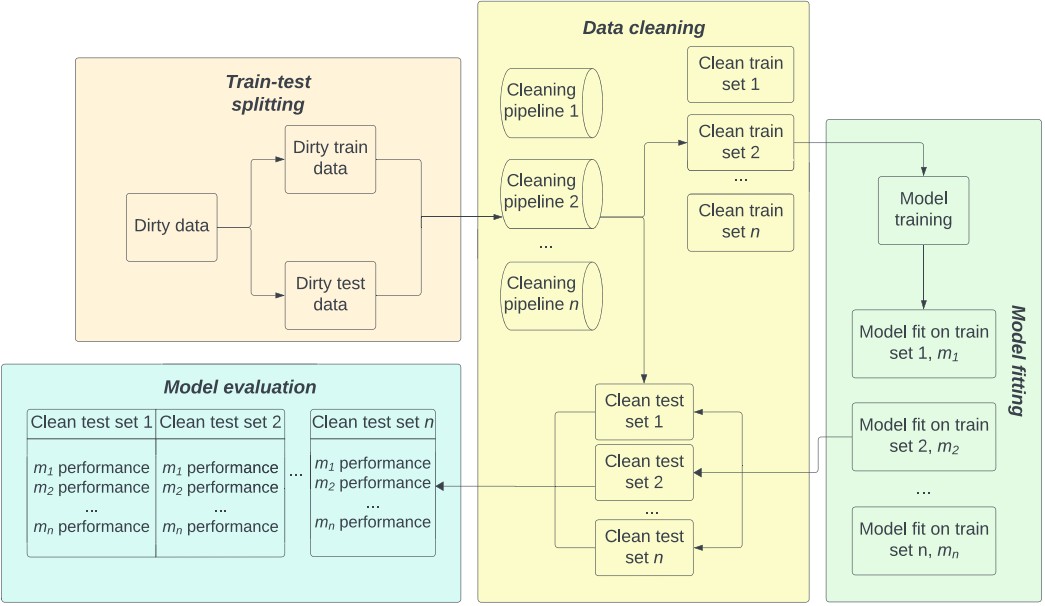

**Figure 1 General experimental procedure for *n* data cleaning pipelines and a single model type.** Both the train and test sets are cleaned using all *n* cleaning pipelines. Models are fitted on all cleaned train sets and inference performed on all cleaned test sets for a total of $n^2$ performance metrics per train-test split. This procedure is repeated 20 times for different train-test splits and performance results averaged. For different model types, the same train-test splits are used.

The USCensus data is only cleaned for categorical missing values. This gives three different cleaning pipelines: 'delete', 'mode', and 'dummy', as described in "Data Cleaning". Hence, per training-test split, nine different models are trained and these nine models are tested on three different test sets, for a total of 27 combinations.

For the Airbnb data, there are 98 possible cleaning pipelines based on all combinations of missing value, outlier, and duplicate detection and correction techniques. Thus, 294 total models have been trained. Each model is tested on the 98 test sets for a total of 28,812 results. This is repeated for 20 different train-test splits and the results are averaged.

Due to the large number of possible cleaning pipelines in the Airbnb data, we will generally refer to them using a number. A table showing the definitions of each cleaning pipeline number is available in the Appendix.

The splitting of the data into train-test splits and subsequent cleaning of both is performed locally. Model training and inference is done on the Netherlands National Supercomputer Snellius (https://www.surf.nl/diensten/snellius-de-nationale-supercomputer). Experiments were run on the 'rome' node, which uses an AMD Rome CPU. Processes were run in parallel across 16 cores with 28 GB memory, and used approximately 9,457 System Billing Units (roughly 591 h total compute time).

## RESULTS

We present the main results of our experiments across the approximately 6,000 total models. As the number of fitted models and performance measurements is so large, it is not possible to present all results here. Instead, we highlight some key findings from the

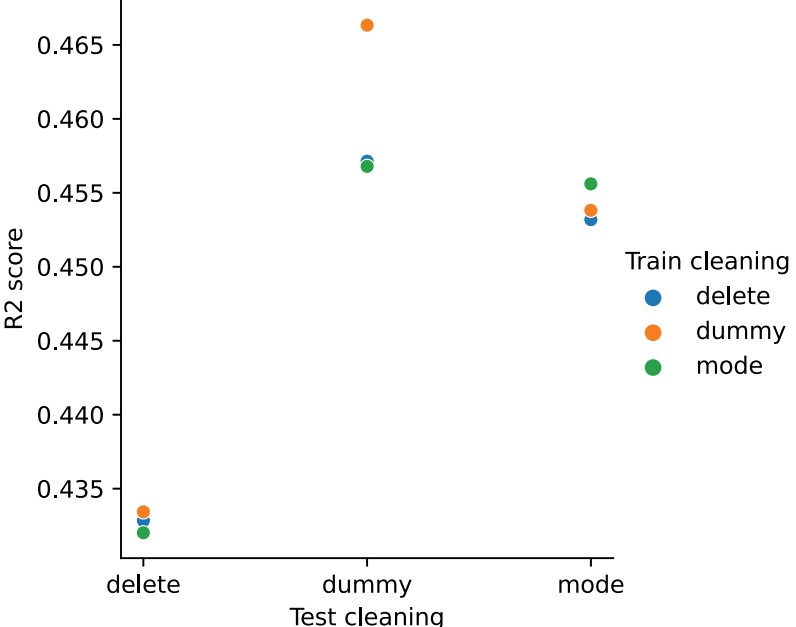

**Figure 2 RFR model performance (USCensus).** The x-axis shows the cleaning pipeline applied to the test data, and the y-axis shows the fitted model performance ($R^2$ score). Different training cleaning pipelines are noted with different colours.          

analysis. To see the full results, the Supplemental Material includes a set of CSV files that includes all the results and a Jupyter notebook that loads and displays them for ease of review. A separate folder contains the code for reproducing the experiments (https:// github.com/jim-g-n/mismatched_data_cleaning).

The findings apply to both datasets. In the following, we provide the evidence for each. Where behaviour across different model types is similar, we only report results for the Random Forest, and note where this is not the case.

### Finding: the method used to clean the test data can have a much greater impact on performance than the method used to clean the training data

*USCensus data*

Figure 2 shows the performance of the RFR models trained on different cleaned training sets when applied to different cleaned test sets for the USCensus data. The x-axis indicates the cleaning pipeline applied to the test sets, the y-axis the $R^2$ score, and the colours differentiate the cleaning applied to the training sets. Results are averaged over different train-test splits.

For a given test cleaning pipeline, there is relatively low variance in the performance based on the training set cleaning pipeline. One exception to this is the test set cleaned by dummy variables, in which case the dummy trained model heavily outperforms the delete and mode trained models.

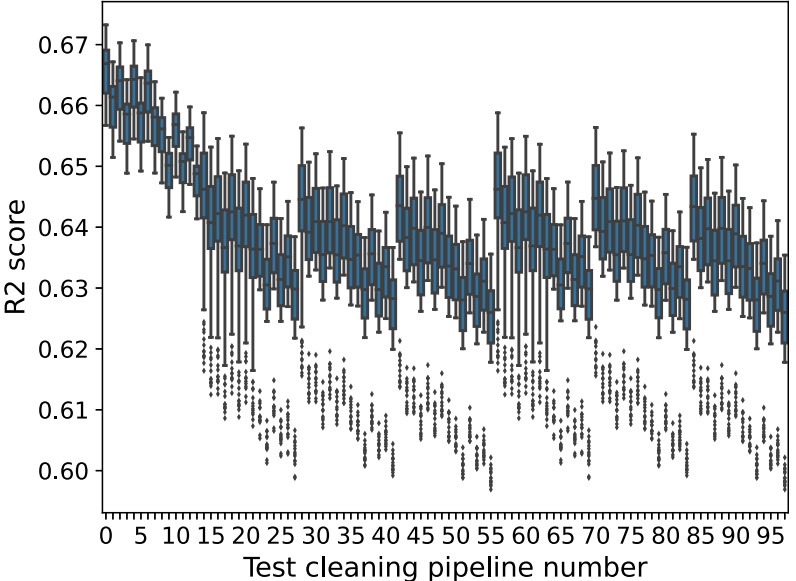

**Figure 3** **RFR model performance boxplots (Airbnb).** The x-axis shows the cleaning pipeline number applied to the test data and the y-axis boxplots of the fitted models' performance ($R^2$ score) over different training cleaning pipelines.

However, the variance between performance for different test cleaning pipelines is much larger, with all models performing far worse on the delete test set, and all models performing best on the dummy test set.

### Airbnb data

Figure 3 shows boxplots of the RFR models' performance for different test set cleaning for the Airbnb data. The x-axis shows the test set cleaning pipeline number, and the y-axis shows the boxplot distributions of the $R^2$ scores for different training set cleaning pipelines on the test set. For a given training cleaning-test cleaning combination, performance is again averaged over the 20 train-test splits.

As in the USCensus data, the variance between test sets can be higher than the variance within test sets. For example, the variance in scores when changing from test set number 0 to test set number 97 is higher than when changing between different training sets within test set 0. There is a noticeable cyclical behaviour in the scores, which corresponds to different data issues being addressed in different ways. For test sets 0–13, missing values are deleted, while for other test sets, they are estimated. Other patterns arise in similar ways.

For this dataset, the differences are more noticeable with a different type of model. Figure 4 shows the same boxplot for the GBR models. In this case, the differences between test sets 0–13 (where missing values are deleted) and test sets 14–97 (where missing values are estimated) are much greater, with all model performances in test sets 14–97 falling below even the minimum model performance on test sets 0–13.

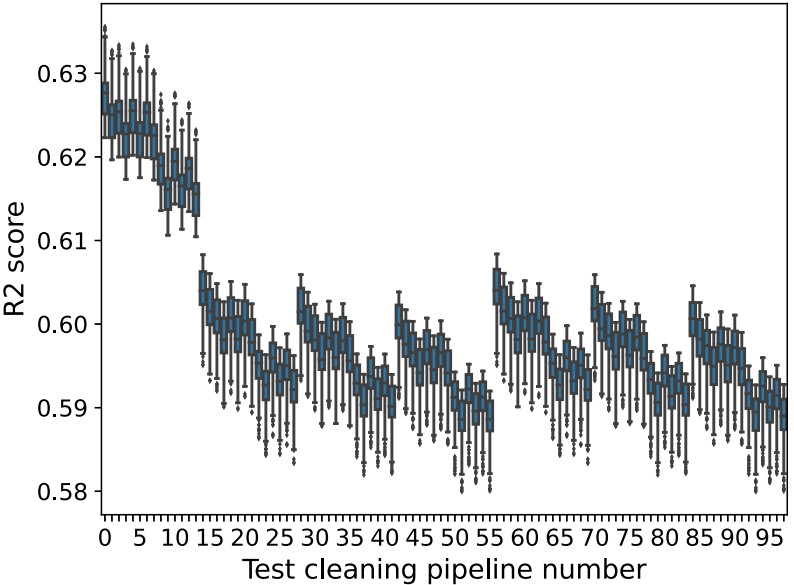

**Figure 4  GBR model performance boxplots (Airbnb).** The x-axis shows the cleaning pipeline number applied to the test data and the y-axis boxplots of the fitted models' performance. The boxplots are created over different training cleaning pipelines.     

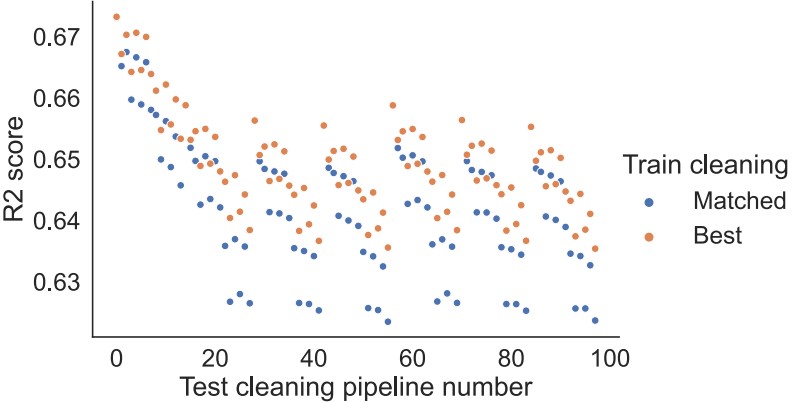

**Figure 5  RFR model matched *vs.* best performance (Airbnb).** The x-axis shows the test cleaning pipeline number and the y-axis the model performance. The blue points indicate average performance over train-test splits when the training cleaning pipeline matches the test cleaning pipeline. The orange points show the maximum average performance when the training cleaning pipeline is allowed to differ from the test cleaning pipeline.     

### Finding: the best performance is not necessarily achieved when the same cleaning process is applied in the test and training data

***USCensus data***

For this finding on the USCensus data, we can again refer to Fig. 2. For the test sets cleaned with dummy or mode, the best-performing models use the same train cleaning (*i.e.*, dummy and mode, respectively). However, for the test sets cleaned with delete, the best-performing model is the one with dummy cleaning on the train sets. However, we note that these performance differences are small and may not be statistically significant.

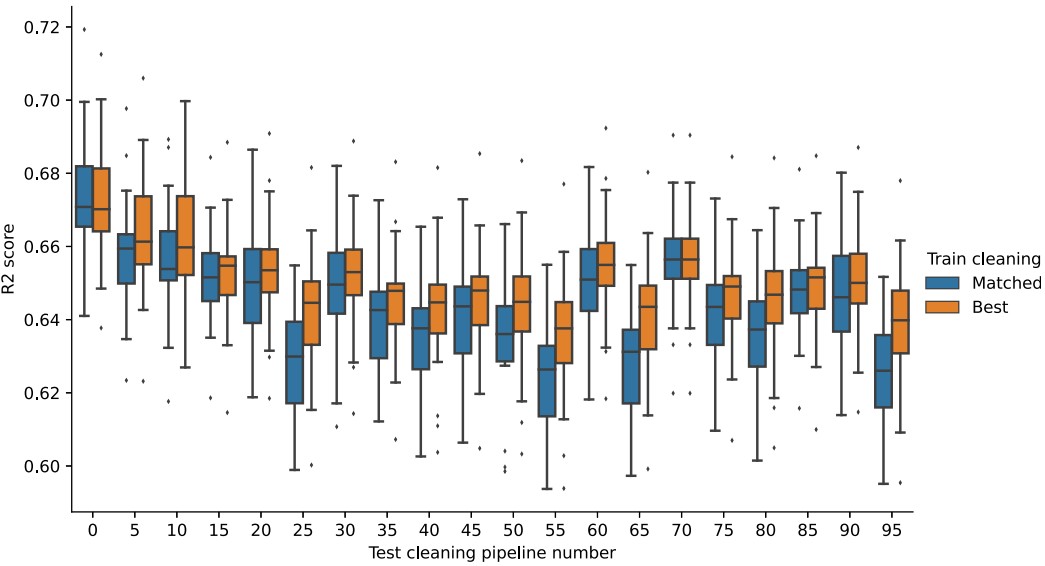

**Figure 6 RFR model matched *vs.* best performance boxplots (Airbnb).** The x-axis shows the test cleaning pipeline number and the y-axis the model performance. The boxplots are created over different train-test splits. The blue boxplots indicate performance when training cleaning matches test cleaning, while the orange boxplots show the maximum performance when training cleaning is allowed to differ from test cleaning. Only the plots for every fifth test cleaning pipeline are shown.

### Airbnb data

Figure 5 compares the performance of RFR models when the training cleaning pipeline matches the test cleaning pipeline, and the best-performing training cleaning pipeline for a given test cleaning pipeline. In most cases (93 out of 98), the training cleaning pipeline that performs best for a given test cleaning pipeline does not match the test cleaning pipeline. This means that for given cleaned test/production data, it is not necessarily best to try to match the cleaning pipeline on one's training data. In this case, there are six unique training cleaning pipelines that perform best on at least one test cleaning pipeline, which cover the range of cleaning options.

Figure 6 shows the boxplots for matched cleaning *vs.* best cleaning model performance when considered over different train-test splits. For ease of visibility, only every 5th test cleaning pipeline is shown. In all cases, there is an overlap in the boxplots between matched *vs.* best for a given test cleaning pipeline, *i.e.*, the best never outperforms the matched in all possible train-test splits. Performing a paired two sample t-test to compare the means over different train-test splits results in 87 of 98 test cleaning pipelines where the best *vs.* matched scores differ significantly at 5% significance (we could expect approximately five false rejects for multiple hypothesis testing).

As with Finding 1, a similar behaviour is observed for the GBR and XGB models, but is less pronounced. In the case of the GBR models, the differences are not statistically significant over different train-test splits (a larger number of train-test sets could change this).

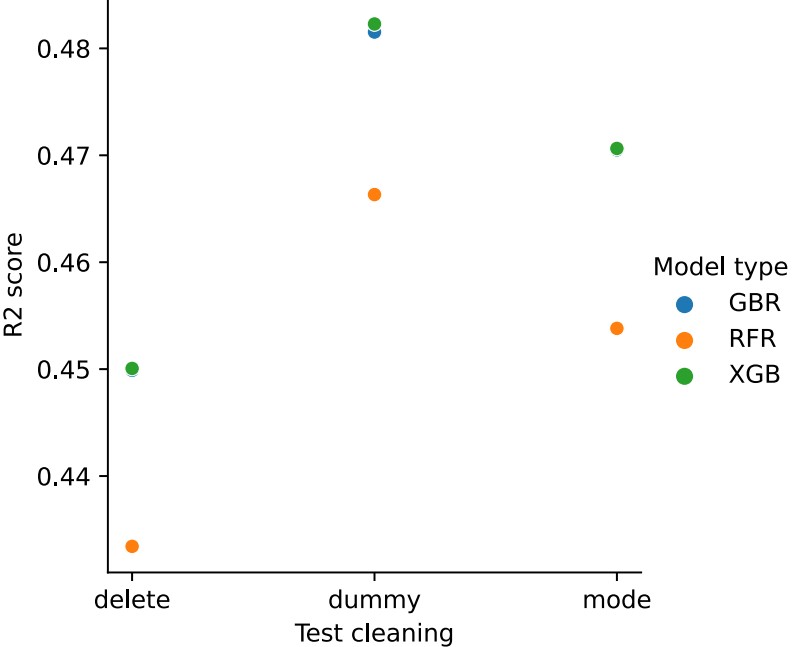

**Figure 7 Different model performance using mode training cleaning (USCensus).** The x-axis shows the test cleaning pipeline, and the y-axis shows the model performance. Different types of model are denoted by different colours.

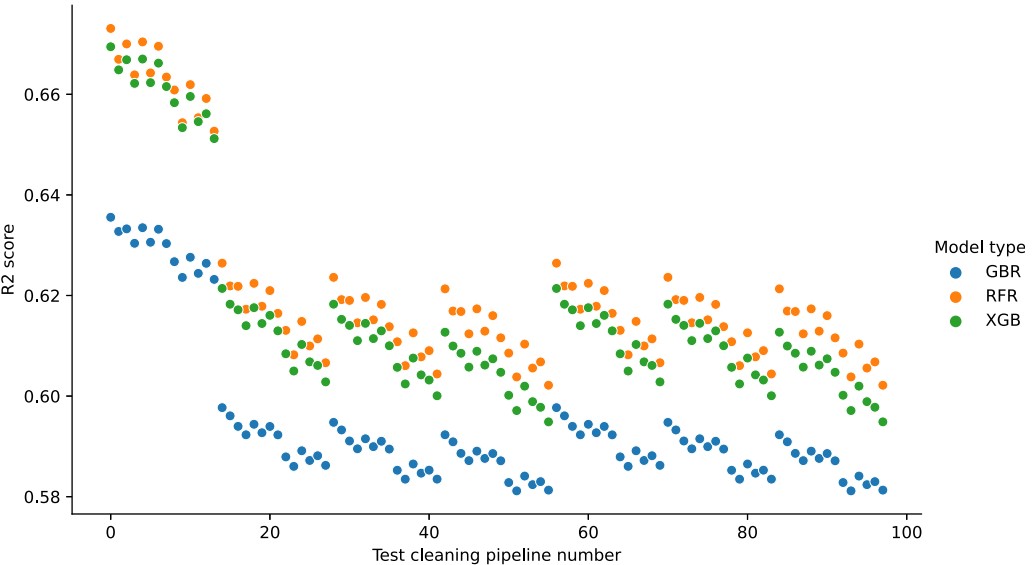

**Figure 8 Different model performance using cleaning pipeline 0 on training data (Airbnb).** The x-axis shows the test cleaning pipeline and the y-axis the model performance. Different model types are noted by colour.

### Finding: different cleaning pipelines for test data can change the best-performing model type for a given training cleaning pipeline

*USCensus data*

Figure 7 shows the relative performance of different types of model trained with the same training cleaning pipeline on different test cleaning pipelines. In these results, all models are trained using 'mode' cleaning.

When using 'mode' cleaning on the training data, the XGB model always performs best when models are compared using the same test cleaning, while RFR always performs worst. However, if the two models are tested on data cleaned using different cleaning pipelines, it can be the case that the RFR model performs best. If 'dummy' test cleaning is used for the RFR model and 'delete' test cleaning is used for GBR, it can appear that RFR is a better choice.

*Airbnb data*

Figure 8 shows the relative performance of different model types when the training data is cleaned using setup number 0. If the model types are compared using the same test cleaning pipeline, RFR always performs best and GBR always performs worst. However, GBR on test cleaning pipelines 0–7 (missing values deleted, outliers detected with standard deviation or ignored) always performs better than both RFR and XGB on test cleaning pipelines 14–97 (missing values estimated).

## DISCUSSION

It is already known that the effects of changing data cleaning pipelines on model performance can be specific to a given model-dataset combination (*Budach et al., 2022*). This means that the findings of these experiments will not necessarily apply/arise in all cases and can be model, dataset, and cleaning pipeline dependent. However, we note that results are consistent between the two datasets tested, and that similar research on comparable problems has led to notable results (*Li et al., 2021*; *Budach et al., 2022*; *Northcutt, Athalye & Mueller, 2021*). With this in mind, we discuss the general findings, the intuitive reasoning behind them, and suggest where such outcomes may arise and how they should be dealt with.

Changing the test set cleaning can have larger impacts on model performance than changing the model itself, either in terms of training set cleaning or model type. A possible explanation for this is that the changed distribution of the test data is significantly easier to model, leading to drastically improved performance for many models. Figure 9 shows the distribution of the cleaned 'Price' variable in the Airbnb data for different cleaning setups covering the range of possible solutions. We see that the level of smoothness in the distributions of this target variable changes based on cleaning setup, which could have a significant impact on model performance. This finding could mean that improving data cleaning pipelines in production could be more effective in maximising performance than retraining and redeploying models, as long as these changes make the cleaned production data more accurately reflect reality.

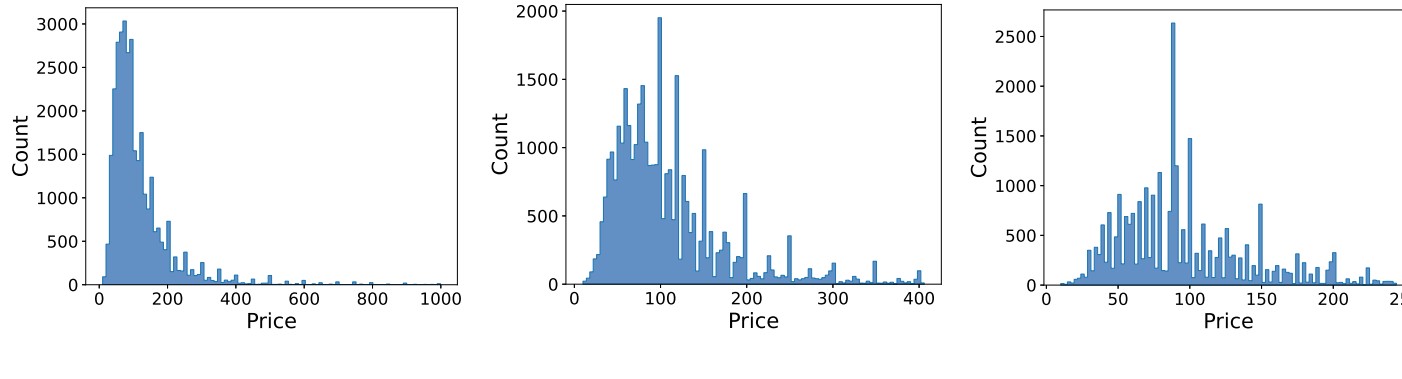

**A:** Cleaning setup 0      **B:** Cleaning setup 16      **C:** Cleaning setup 39

**Figure 9** Distributions of cleaned target variable ('Price') in Airbnb data using cleaning setup: (A) number 0, where missing values are deleted, and outliers and duplicates are unaddressed; (B) number 16, where missing values are replaced with the mean, outliers are detected using SD and replaced with the mean, and duplicates unaddressed; (C) number 39, where missing values are replaced with the median, outliers are detected with IQR and replaced with the median, and duplicates are deleted. 

**Table 4** Top 5 training pipelines based on different criteria (Airbnb).

| 'Price' distribution | Multiple variable distribution | RFR performance | GBR performance | XGB performance |
|---|---|---|---|---|
| 73 | 28 | 42 | 56 | 56 |
| 31 | 70 | 84 | 15 | 14 |
| 29 | 30 | 70 | 14 | 42 |
| 71 | 72 | 28 | 28 | 84 |
| 47 | 60 | 14 | 85 | 28 |

It can also be tempting to try to match training cleaning and production data cleaning as closely as possible. However, these results show that certain training cleaning approaches can be highly effective across multiple different test cleaning pipelines. In many cases, the best-performing training cleaning pipeline is different from the test cleaning pipeline. As each cleaning decision relies on some sort of assumption, it could be that the assumptions of certain pipelines are less restrictive than others, leading to models that extend better to other conditions. However, predicting which cleaning pipeline and model combinations will perform best in advance can be difficult.

As an example, Table 4 shows the top five best performing training cleaning pipelines in the Airbnb data based on different criteria. For the first two columns, we show the training cleaning pipelines whose distributions most closely match those in the test data averaged over all test cleaning pipelines. In order to quantify similarity in distributions, we calculate the Jensen Shannon distance based on histograms with 100 bins. This is a relatively crude approach and can be sensitive to choice of number of bins, but gives a good initial impression. The 'Price' column uses only the distribution of the target variable, while the multiple variable column uses the target plus two representative features, one specific to the particular Airbnb rental (number of reviews) and one for the region of the Airbnb (cost of living index). We note that the best performing training cleaning pipelines can differ

depending on which criteria is used, and one cannot simply focus on cleaning to match distributions.

Another difference we see is in the number of unique training cleaning pipelines performing best for a given test cleaning pipeline. When analysing the RFR models, there were six unique training cleaning pipelines that performed best on at least one test cleaning pipeline. When considering distributions instead, there are 22 training cleaning pipelines that most similarly match the 'Price' distribution on at least one test cleaning pipeline. In other words, it is potentially more likely that a cleaning pipeline could perform well in terms of distribution matching than in terms of downstream model performance.

All of this suggests that there is an interaction between the models and training cleaning pipelines, and using only distributions for selecting cleaning pipelines may miss these interactions. One possible explanation for this is that some models have built-in tools for dealing with data quality issues, and perhaps perform better when using poorly cleaned data. This could especially be true when using deep learning models, which theoretically deal well with capturing complex distributions. However, such models are not immune to data quality issues, remaining dependent to some extent on the cleaning process. This is especially true when it comes to robustness and fairness, which are active areas of research (*Whang & Lee, 2020*; *Whang et al., 2023*). Furthermore, these models are expensive and time-consuming to train, or are pre-trained (*Han et al., 2021*; *Papernot et al., 2017*), meaning that they will still need to deal with mismatches in training and production data cleaning pipelines over time. It is also possible that these simpler models we have tested could be more robust and less at risk of overfitting. *Northcutt, Athalye & Mueller (2021)* found that lower capacity models could perform better than higher capacity ones after correcting errors on test data; likewise, *Budach et al. (2022)* found that in some cases simpler models were more robust to data quality issues (but this was dependent on the dataset). Considering all these possibilities, there is room for performing further analysis as in this article to ascertain to what extent our findings apply to more sophisticated models.

The final takeaway from these findings is the need for transparency in data cleaning and the evaluation of ML models. Results using different cleaning pipelines should be reported, as adjusting these can have significant impacts on performance. This could in some cases be used to make results appear better than they are, especially when making non-like-for-like comparisons to other models/cleaning pipelines. In general, performance differences of state-of-the-art models on benchmark datasets are often very small. For example, the percentage correct for the top five models on the CIFAR-10 image classification task (*Krizhevsky & Hinton, 2009*) varies between 99.42% and 99.61% (https://paperswithcode.com/sota/image-classification-on-cifar-10, accessed September 2024); for the object detection task on COCO minival (*Lin et al., 2014*), the box AP scores of the top five models vary between 64.6% and 65.9% (https://paperswithcode.com/sota/object-detection-on-coco-minival, accessed September 2024). However, these benchmark datasets would have gone through a cleaning pipeline themselves, which could affect the relative performance of different models. This need for transparency could also be present in fairness issues for such problems. For example, the USCensus dataset tested has a number of potentially

**Table 5 Race counts (USCensus).**

| Race | Full dataset | Missing values subset |
|---|---|---|
| White | 27,816 (85.4%) | 1,883 (78.5%) |
| Black | 3,124 (9.6%) | 307 (12.8%) |
| Asian/Pacific Islander | 1,039 (3.2%) | 144 (6%) |
| American Indian/Inuit | 311 (1%) | 40 (1.7%) |
| Other | 271 (0.8%) | 25 (1%) |

sensitive variables (Race, Sex, Native Country). Table 5 shows the counts of different races over the full dataset and the subset of rows with at least one missing value. Shown in brackets is the percentage. We note that minorities are more likely to have missing values, and they will thus have data cleaning applied disproportionately. As a result, mismatches in data cleaning pipelines could have a larger effect on minorities. Understanding these effects could be crucial in ensuring fairness.

## Suggestions for implementation

The results of this study show the need for carefully considering how production data may change over time when fitting and selecting an ML model. Testing candidate models on data with varying cleaning pipelines could provide a more holistic view on the models' robustness to changing production conditions. Models can be selected based on their overall performance across many cleaning pipelines. The approach taken in this article of testing all possible combinations of training and test cleaning would be less feasible with larger datasets and more complex/expensive models. Instead, a curated set of candidate cleaning pipelines can be created that lead to a comprehensive range of distributions in the data. Models that perform well across these pipelines are likely to be more robust to future changes. However, there is additional complexity in the interaction of model performance and cleaning pipelines that should be carefully considered.

It is also necessary to monitor how a model's performance changes in deployment as cleaning pipelines change. There is already a wealth of research on this topic. Approaches such as those by *Mahadevan & Mathioudakis (2024)* and *Dong et al. (2024)* consider how shifts in the production data can affect ML model performance and suggest when retraining of the model will be necessary. Such approaches generally rely on measuring changes in important attributes for a given model. Less sophisticated approaches simply consider how shifts affect inference scores (*EvidentlyAI, 2024*). As shown through our experiments, changes could in fact increase inference scores in some cases, emphasising that retraining is not always necessary.

## CONCLUSIONS

With the increasing amount of data reuse and the known effects that data cleaning can have on ML models and their performance, the need to test and quantify different data cleaning pipelines is apparent. The experiments performed in this article showed some

unexpected findings, such as the potential advantages in the performance of the ML model when training and test set cleaning pipelines are mismatched. The choice of models can also be affected by the cleaning pipeline applied to test data, which may arise particularly when different groups are testing on the same dataset. We also see that model performance can vary more through changing test set cleaning pipelines than training set cleaning pipelines.

Taking this into account, it could be more prudent to select models based on their performance on a range of different test cleaning pipelines, as this would hopefully make them more robust if changes are made to production data cleaning in the future. Accurately reporting on multiple cleaning pipelines when evaluating new models can also increase confidence in the wide-ranging applicability of a model. Such efforts are probably less necessary in the case where it is possible or inexpensive to retrain models.

There are some limitations to this work and avenues for future extensions. As mentioned in *Li et al. (2021)* and *Neutatz et al. (2022)*, there is a limited number of real-world datasets with dirty, non-synthetic, and impactful errors plus gold standard cleaned versions. As a result, our analysis has been limited to only two of these datasets. While the results are consistent across these two datasets, further experiments are necessary to generalise all findings. That being said, there are some intuitive explanations as to why we might expect these results to generalise. Also worth noting is that we tested a relatively small set of moderately complex regression models. As this problem and approach are relatively new, focussing on simpler, under-studied regression models served a good initial foundation, but further testing using more complex models or other types of models will garner deeper insights.

## ACKNOWLEDGEMENTS

We thank SURF (www.surf.nl) for support in using the National Supercomputer Snellius.

### Funding
The authors received no funding for this work.

### Competing Interests
The authors declare that they have no competing interests.

### Author Contributions
- James Nevin conceived and designed the experiments, performed the experiments, analyzed the data, performed the computation work, prepared figures and/or tables, authored or reviewed drafts of the article, and approved the final draft.
- Michael Lees conceived and designed the experiments, authored or reviewed drafts of the article, and approved the final draft.
- Paul Groth conceived and designed the experiments, authored or reviewed drafts of the article, and approved the final draft.

## Data Availability

All code and results are available at Zenodo: jim-g-n. (2025). jim-g-n/ mismatched_data_cleaning: final (final). Zenodo. https://doi.org/10.5281/zenodo. 14945262. The 'analysis' folder contains results as CSV files, two Jupyter notebooks for performing analysis, and an appendix PDF. The 'code' folder contains code for reproducing the experiments, including all necessary raw data.

The data is available from CleanML at GitHub: https://github.com/chu-data-lab/ CleanML. The original data was downloaded from https://www.dropbox.com/s/ nerfrhbrseev928/CleanML-datasets-2020.zip?dl=0.

## Supplemental Information

Supplemental information for this article can be found online at http://dx.doi.org/10.7717/ peerj-cs.2793#supplemental-information.

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
