# Peer review of "The effects of mismatched train and test data cleaning pipelines on regression models: lessons for practice"

_PeerJ Computer Science, doi:10.7717/peerj-cs.2793_

## Round 0.1 · original submission · Major Revisions

Three reviewers have provided careful insights about the manuscript. Overall, they expressed the value of the work described in the manuscript. However, they had several suggestions for improving the manuscript, and they sometimes agreed about particular aspects that should be improved. Two reviewers suggested that including a classification analysis (with a traditional algorithm and a deep-learning algorithm) would improve the generalizability of your findings. I agree, but I do not want to go as far as mandating additional experiments. If you do not, you should clearly provide a strong justification for not including those in the paper. Other suggestions focus mostly on adding details to the writing or improving the writing. Please address those carefully. I look forward to your revision.

Reviewer 1 ·

Basic reporting

The article effectively presents the importance of the issue, particularly regarding the growing significance of machine learning (ML) in various fields. However, it could benefit from some improvements.
Firstly, consider expanding the discussion on the costs of retraining models, including detailed aspects related to time and resources. This will help readers better understand the challenges associated with model management. Additionally, incorporating specific examples from industry would emphasize the practical significance of the topic.
While the focus on regression studies is commendable, a brief overview of other types of ML tasks that may also be affected would enrich the context. Finally, instead of concluding the introduction with experiments, adding a section that underscores how the work contributes to the current state of knowledge in the field would enhance clarity.

Experimental design

The experimental design section could be improved by addressing a few key points. It is essential to clarify the difference between studies related to classification and the focus on regression. Explaining why this distinction matters and how metric differences impact the research would add depth to the study.
Expanding the justification for the selection of regression models (RFR, GBR, XGB) by referencing results from previous studies that support this choice, particularly concerning low-quality data, would strengthen the rationale behind the experimental design. In section 3.1, adding comments about the impact of various cleaning methods on results would be beneficial. For instance, explaining how the removal of rows with missing values affects modeling would provide more insight.
Furthermore, a more detailed justification for the selected strategies to handle missing data and outliers would enhance the robustness of the methodology. While using real-world datasets (USCensus and Airbnb) is commendable, explaining why these specific datasets are suitable for studying the impact of data cleaning would provide valuable context.

Validity of the findings

The validity of the findings could be enhanced by addressing several areas. The discovery that the best results do not always come from the same data cleaning methods for training and test sets is significant. However, discussing practical ways to handle differences in data cleaning approaches would add to the validity of the findings.
Moreover, emphasizing that changes in the cleaning of test data may have a greater impact on results than merely changing the model is crucial. It would be useful to indicate what should be prioritized in real applications: improving production data cleaning or retraining models.

Additional comments

It would be helpful to clarify how cleaning practices during evaluation align with industry standards, such as the independent cleaning of training and test sets. The article mentions the use of the supercomputer Snellius for model training, which is valuable. However, detailing the resources required for these computations, such as computing hours and GPU/CPU usage, would provide additional insight into the scale of the experiments.
Lastly, given the complexity illustrated by the numerous possible cleaning pipelines, discussing how different cleaning strategies can lead to vastly different results in predictive models would effectively reflect real-world challenges.

·

Basic reporting

Pipeline Mismatch Impact Analysis: The study highlights that mismatches between training and test data cleaning pipelines can influence model performance, but it lacks depth in exploring why certain mismatches improve performance. Understanding whether the improvements arise from data-driven corrections or fortuitous circumstances could help practitioners make more informed choices about when and how to apply pipeline updates in real-world scenarios.

Scope of Model Types: Although the study evaluates over 6000 models, it does not specify the diversity of regression models tested. Certain models may be more robust to data inconsistencies or have built-in mechanisms to handle noise, which could influence the observed performance variations. A detailed breakdown of how different model architectures respond to pipeline mismatches could provide useful insights into model robustness under varying data conditions.

Generalization and Real-World Applicability: The findings indicate that data cleaning pipeline changes can affect performance, potentially even improving it. However, this improvement may not generalize well across different datasets or domains. The study could be strengthened by testing across multiple datasets with varied data quality issues (e.g., missing values, outliers, duplicates) to see if the observed trends hold universally or are dataset-specific.

Experimental design

Practical Implications for Model Retraining: The study suggests that frequent updates to cleaning pipelines without model retraining are common. However, the paper does not address the operational challenges or feasibility of implementing continuous retraining to accommodate data pipeline changes, which would be beneficial in practical settings. Discussing a framework for deciding when retraining is necessary based on pipeline modifications would add practical value.

Bias and Fairness Considerations: When data cleaning steps are updated inconsistently, especially if training data remains “cleaned” differently than production data, this could unintentionally introduce bias. The study would benefit from an analysis of whether pipeline mismatches disproportionately affect specific data subsets or contribute to fairness issues, particularly if used in applications where equitable performance is crucial.

Validity of the findings

Impact on Model Selection: The study mentions that pipeline mismatches may influence model selection, yet it does not explore the potential risks involved. For example, selecting a model that performs well on mismatched data might not generalize as effectively as one that performs robustly across various cleaning pipelines. Providing recommendations or strategies for model selection under uncertain cleaning conditions could be an actionable outcome for practitioners.

Reviewer 3 ·

Basic reporting

See detailed comments

Experimental design

See detailed comments

Validity of the findings

See detailed comments

Additional comments

Summary
The impact of data quality on machine learning has become a critical focus area within the database and ML communities, encapsulated under the concept of "data cleaning for AI”. In the manuscript’s purview, data quality issues are addressed through data cleaning pipelines that process and refine datasets. Ideally, these pipelines are applied consistently to both training and test datasets. However, the authors highlight that in practice these pipelines often evolve over time. When different versions of these pipelines are applied separately to the training and test datasets, it results in discrepancies in data distribution and quality.

The authors conducted a series of quantitative experiments to evaluate how variations in data cleaning pipelines influence the predictive accuracy of several regression models. Their study employs three regression models (Random Forest Regression, Gradient Boosting Regression, XGBoost Regressor) and utilizes two widely recognized datasets. The analysis focuses on three types of data errors: missing values, outliers, and duplicates. Specific detection and correction strategies were implemented for each error type, forming distinct data cleaning pipelines.

To assess the effect of mismatched cleaning pipelines, the authors applied all possible combinations of these pipelines to the training and test datasets. The impact on the performance of the regression models was then analyzed for each case. For each dataset and cleaning pipeline combination, 20 train-test splits were examined to ensure robustness in the findings.

The authors employed various visualizations to illustrate the effects of mismatched cleaning pipelines across the two datasets. Their findings reveal that it is more important to provide a good cleaning pipeline for the test data than for the training data. Furthermore, they conclude that aligning the cleaning pipelines for training and test datasets does not always lead to improved prediction accuracy, challenging common assumptions about data preprocessing practices.

Strong Points
- The authors have identified a significant and relevant topic, addressing it through a well-designed experimental setup. Evaluating the cross-product of all possible cleaning pipeline combinations is a logical approach, providing a comprehensive perspective on the various conditions and their effects.
- The results are well-structured as “Findings” and supported by clear, visually engaging figures. The authors address various interesting points in the findings to answer the initial question of the impact of mismatched cleaning pipelines from different perspectives.
- The representation of the experimental pipeline (Figure 1) is well-designed.

Major Issues
- The authors limit their experiments to three fundamental machine learning algorithms, two of which are tree-based. While they acknowledge this limited selection as an opportunity for "further analysis”, I recommend expanding the scope by including at least one representative algorithm from regression-based models (e.g., Support Vector Machines) and one from deep learning models (e.g., Multi-Layer Perceptrons). This broader inclusion would enhance the generalizability and robustness of their findings.
- The use of only two datasets is insufficient to generalize the results. The authors argue that "there is a limited number of real-world datasets with dirty, non-synthetic, and impactful errors plus gold standard cleaned versions". However, based on my understanding of the authors' description in the manuscript (“slightly altered version of the MVCleaner class from CleanML”) and provided code, they implemented their own cleaning procedures/pipelines, effectively creating their own cleaned versions of the datasets.
Moreover, CleanML provides access additional datasets beyond the two used in this study. Even if the authors relied on externally cleaned versions, which seems not to be the case, those datasets would strengthen their analysis. To address these limitations, I recommend incorporating at least one larger dataset with a minimum of one million rows. This addition would further enhance the robustness and generalizability of their findings.
- The authors should provide explanations or, at the very least, intuitive reasoning behind their findings. At present, the results are largely descriptive, merely reiterating what is apparent from the figures. Adding a deeper analysis or interpretation would significantly enhance the value of the findings and offer readers a clearer understanding of the underlying patterns and implications.
- “Both the training and test sets are cleaned independently using n cleaning pipelines. This results in n cleaned training sets and n cleaned test sets” -> The authors are intentionally producing a covariate shift. Dong et al. (https://dl.acm.org/doi/pdf/10.14778/3681954.3681984) also considered covariate shifts and tried to identify portions of the data that are harmful to the downstream model. The authors should relate their work to the work of Dong et al. .

Minor Issues
- Spelling inconsistencies: "dataset" vs. "data set"
- Inconsistencies in use of "machine learning". The authors switch between “machine learning” and “ML”, even after introducing the abbreviation.
- The authors should differentiate their work from AutoML systems and discuss whether existing AutoML systems address the issue of cleaning pipeline mismatches highlighted in this study. Furthermore, they should provide an overview of how future AutoML systems could incorporate the findings of this manuscript to improve their handling of data preprocessing and ensure consistency between training and test datasets.
- Change “This information is summarised in Table 1.” to “This cleaning information is summarised in Table 1.”
- “Numerical outliers can be ignored or identified and corrected.” -> Clarify what is meant by ignored outliers. Ignoring an error should not be part of a cleaning pipeline; the same applies to duplicates. Alternatively, if this approach is intended to simulate an undetected error, then "ignore" should also be treated as an identification technique for missing values to ensure consistency across error types.
- Section 3.2 should come before the current Section 3.1., as data cleaning relies logically on actual data.
- Clarify in Table 1 that for the identification technique of missing values, "NA" constitutes a placeholder.
- “For each dataset and model combination, the grid search parameters are chosen on the basis of the full data set with missing values deleted and no other cleaning applied.” -> Why were missing values deleted? The more realistic scenario would be to conduct the parameter search on the full dirty dataset, as this is the one that is available at the beginning. While I understand that performing a parameter search after each cleaning procedure would be unrealistic (as it is costly), conducting the parameter search on the fully dirty dataset would provide a more accurate simulation of real-world scenarios.
- In the context of Section 4.3.1, considering all possible combinations of cleaning pipelines would result in nine configurations. Including all these configurations in Figure 7 would provide a more comprehensive and detailed presentation of the results.
- The provided code is not easily runnable: The aggregation functions .mean() and .std() can only be applied to numerical columns. Thus, the function call needs the additional parameter: numeric_only=True -> .mean(numeric_only=True). After adding this, it was runnable.
- “In most cases, the training cleaning pipeline that performs best for a given test cleaning pipeline does not match the test cleaning pipeline.” -> The authors should clarify how often the cleaning pipelines actually match. This is also important for the discussion around Figure 6.
- In Section 4.2.2, nearly every sentence appears in a separate paragraph, which disrupts the flow of the text. The authors should reconsider the organization and grouping of related ideas to create more cohesive and well-structured paragraphs. This applies to the whole manuscript.
- In the context of Figure 2, clarify whether the presented results are averaged over the 20 train-test splits.
- Figures are rather large and thus far away from the place where they are referenced. Reducing their size (but not their font-size!) could help.
- Figures that are not included in the manuscript could be shown in the appendix, rather than only being generated through the Python code.

---

## Round 0.2 · Minor Revisions

Thank you for addressing the reviewers' comments. One reviewer was content with the changes. Another reviewer felt that their comments had not been fully addressed. I tend to agree that additional work would improve the paper. I have also added some minor requests. Below are more details.

* Please address the following comment from the first round of reviews. Or if I missed it, please indicate how it was addressed. "The study would benefit from an analysis of whether pipeline mismatches disproportionately affect specific data subsets or contribute to fairness issues, particularly if used in applications where equitable performance is crucial."
* A reviewer indicated: "It would be insightful to quantify discrepancies between the training and test distributions and relate them to the reported results. I suggest using a measure like the Kullback-Leibler divergence to assess these discrepancies...Additionally, to what extent can the test data distribution deviate from the training distribution while still yielding reliable results? it is insufficient to only relate results to the pipeline; the discussion should be extended to account for differences in the distributions." Please address this with additional analysis and/or with specific commentary in the paper.
* A reviewer indicated, "Another interesting perspective is whether applying cleaning pipeline A to the training data and cleaning pipeline B to the test data could reduce discrepancies in the distributions and thereby improve prediction accuracy. Therefore, it is insufficient to only relate results to the pipeline; the discussion should be extended to account for differences in the distributions." Please address this with additional analysis and/or with specific commentary in the paper.
* A reviewer indicated, "In this context, more sophisticated models should be considered to analyze whether deep learning models can better generalize from distributions, even in cases where the training data follows a distribution like Figure 9c and the test data follows a distribution like Figure 9a." The manuscript mentions that relatively simple algorithms have been used. But please address this point more specifically.
* Please cite the sklearn software and provide version number(s) that you used.
* Thank you for providing a GitHub site. Please add more detail to the README file so that readers know exactly which Python packages to install, including specific version numbers for these packages. Indicate what commands a reader would use to open the notebooks and in what order.
* Please also address the minor points that the reviewer provided.

Please provide a point-by-point response to each of these comments.

Reviewer 1 ·

Basic reporting

The authors have successfully revised the manuscript in line with the suggestions provided earlier. In its current form, the article is suitable for publication.

Experimental design

The authors have successfully revised the manuscript in line with the suggestions provided earlier. In its current form, the article is suitable for publication.

Validity of the findings

The authors have successfully revised the manuscript in line with the suggestions provided earlier. In its current form, the article is suitable for publication.

Additional comments

The authors have successfully revised the manuscript in line with the suggestions provided earlier. In its current form, the article is suitable for publication.

Reviewer 3 ·

Basic reporting

See below

Experimental design

See below

Validity of the findings

See below

Additional comments

I would like to thank the authors for their revision. They have addressed most of the points I previously raised, leading to an overall improvement of the manuscript. However, there are still several important issues that should have been or need to be addressed:

- Figure 9 is a valuable addition, but it raises critical questions. The manuscript refers to "reality," but from a model perspective, this reality is defined by the training data. It would be insightful to quantify discrepancies between the training and test distributions and relate them to the reported results. I suggest using a measure like the Kullback-Leibler divergence to assess these discrepancies.
The key question is how different cleaning pipelines affect the distributions of both training and test data, and how well the ML model generalizes from the training distribution. Additionally, to what extent can the test data distribution deviate from the training distribution while still yielding reliable results?
In this context, more sophisticated models should be considered to analyze whether deep learning models can better generalize from distributions, even in cases where the training data follows a distribution like Figure 9c and the test data follows a distribution like Figure 9a.
Another interesting perspective is whether applying cleaning pipeline A to the training data and cleaning pipeline B to the test data could reduce discrepancies in the distributions and thereby improve prediction accuracy.
Therefore, it is insufficient to only relate results to the pipeline; the discussion should be extended to account for differences in the distributions.

- "for which the effects of data quality/data cleaning are generally less well-understood than for classification tasks." -> This statement should be supported with references that highlight the predominant focus on classification in prior research.

- For all figures that present test cleaning pipelines in the form of pipeline numbers on the x-axis, I recommend adding additional labels to indicate the intervals for the different cleaning methods.

- It is a bit of a shame that the authors decided not to pursue further (different and larger) datasets. Even the first sentence of the abstract mentions "large-scale" datasets. As it stands, the experiments still involve only two (rather small) datasets, diminishing the generalizability of the results and insights. While I understand the difficulty of obtaining such datasets as laid out in the new Section 3.1, the weakness remains. One solution might be to use synthetically polluted data. Another might be to actually find or create more real-world examples.

---

## Round 0.3 · Minor Revisions

Thank you for addressing the reviewers' comments. The manuscript is nearly ready for publication. I just have one minor tweak to request. On line 393 (of the tracked changes version), please change "who’s" to "whose".

---

## Round 0.4 · accepted · Accept

Thank you for addressing the requested changes.